

# One hundred years of atmospheric and marine observations at Utö Island, the Baltic Sea

Lauri Laakso[1,2], Santtu Mikkonen[3], Achim Drebs[1], Anu Karjalainen[1], Pentti Pirinen[1], and Pekka Alenius[1]

[1]Finnish Meteorological Institute, Erik Palménin aukio 1, FI-00560 Helsinki, Finland
[2]North-West University, Unit for Environmental Sciences and Management, Potchefstroom, South Africa
[3]University of Eastern Finland, Kuopio, Finland

*Correspondence to:* Lauri Laakso (lauri.laakso@fmi.fi)

**Abstract.** Utö Atmospheric and Marine Research Station is located on Utö Island (59º46'50 N, 21º22'23 E) at the outer edge of the Archipelago Sea, Baltic Sea towards the Baltic Sea Proper. Meteorological observations at the island started in 1881 and vertical profiling of sea water temperature and salinity in 1900. In this study, we analyze long-term changes of atmospheric temperature, cloudiness and sea salinity, temperature and ice cover. Our main dataset consists of 248367 atmospheric temperature observations, 1632 quality assured vertical seawater temperature and salinity profiles and 8565 ice maps, partly digitized for this project. We also use North Atlantic Oscillation (NAO) and Major Baltic Inflow (MBI) data from the literature as reference variables to our data. Our analysis is based on statistical method utilizing dynamic linear model. The results show an increase in the atmospheric temperature at Utö, but the increase is significantly smaller than on land areas and takes place only since early 1980's, with a rate of 0.4 °C/decade during the last 35 years. We also see an increase on sea water temperatures, especially on the surface, with an increase of 0.3 °C/decade for the last 100 years. In deeper water layers the increase is smaller and influenced by vertical mixing, which is modulated by inflow of saline water from the North Sea and fresh water inflow. Date when air temperature in the spring exceeds +5 °C has become 5 days earlier from the period 1951-1980 to period 1981-2010 and date when sea surface water temperature exceed +4 °C has changed 9 days earlier. Sea ice cover duration at Utö shows a decrease of approximately 50% during the last 35 years. Based on the combined results, it is possible that the climate at Utö may have changed into a new phase, in which ice do not reduce the local effects of global temperature increase.



## 1 Introduction

Recently, average atmospheric concentration of carbon dioxide has exceeded 400 ppm (Kilkki et al., 2015) and the effects of climate change are getting continuously more visible throughout the Earth (IPCC, 2013; Mikkonen et al., 2015; Iles and Hegerl, 2017). The Baltic Sea, with shallow waters and variable ice cover rapidly responses to both annual and long-term

changes (Lehmann et al., 2011; HELCOM, 2013). However, the responses are still slower than those observed over land areas, due to thermal inertia of the water body.

Previous studies from the Baltic Sea area show that during the 20th century, air temperatures have increased until 1930, decreased until 1960's and started to increase again since 1980's (HELCOM, 2013). Simultaneously, the sea surface temperatures have followed the atmospheric temperatures, with a clear increase due to recent decrease in duration of ice cover. Sea water

salinities at the Baltic Sea follow both changes in fresh water inflow and Major Baltic Inflows (MBI). The MBI's increase stratification, leading to reduced vertical mixing.

In Finland, the longest observed data series containing sea temperatures and salinities together with meteorological variables are from the Island of Utö at the outer edge of the Archipelago Sea (Ahlnäs, 1961). Recently, Utö station was selected as one of the WMO long-term observing stations in the recognition of irreplaceable cultural and scientific heritage (World Meteoro-

logical Organization, 2017). However, despite observations having started already in 1881, limited number of meteorological studies (e.g., Riihelä et al., 2015; Laapas and Venäläinen, 2017) and only few studies focusing on sea ice and hydrography (Ahlnäs, 1961; Haapala and Alenius, 1994; Haapala and Lepparanta, 1997; Jevrejeva et al., 2004) have been published. One reason for this is that significant part of the observation data has not been digitized or quality assured until the current study. One of the aims of our paper is to fill this gap and analyze these time series for getting information on typical atmospheric and

marine conditions and ranges of variability at Utö. Another aim is to support and provide site characterization and background information for the future studies focusing on combined physical, chemical and biological processes at the recently developed Utö Atmospheric and Marine Research Station.

In this paper, we investigate meteorological, hydrographic and sea ice observations carried out in Utö, Baltic Sea for the period 1881-2016. First, we describe the general environmental characteristics of the measurement place. Next, we continue

by describing the observations, data quality assurance methods, and the tools used for statistical analysis of the data. The paper continues with time series of air temperature, cloudiness and seawater temperatures, salinities and densities. Paper is closed with conclusions, while short description of current observations at Utö research station is left as an Appendix.

## 2 Measurement site and general characteristics

The Baltic Sea is a shallow semi-enclosed seasonal sea with average depth of 55 meters and maximum depth of 459 m

(Lepparanta and Myrberg, 2009). In spite of the shallow depth, the vertical stratification is strong in summer in shallow areas and throughout the year in areas that are deeper than the mean depth. The upper layer of the sea has a strong seasonal cycle, which is also reflected partly to the deeper water. Parts of the Baltic Sea are ice covered every winter, so that the extent of the annual maximum ice cover varies among $50 \cdot 10^3$ and $340 \cdot 10^3$ km$^2$ (Seinä and Palosuo, 1996; Vainio, 2001) of the total area



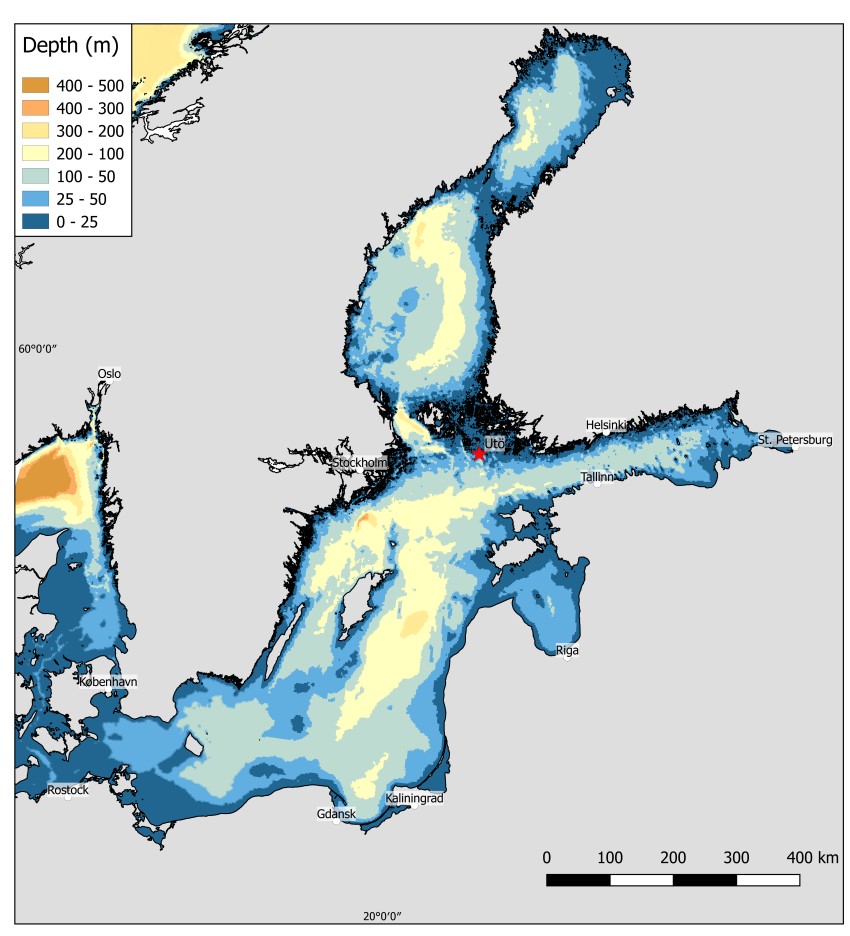

**Figure 1.** Map of Baltic Sea. Utö measurement site is indicated with a red star.



of $420·10^3$ km$^2$. The salinity varies from more than 20 ‰ in Kattegat down to less than 2 ‰ at the ends of the large bays in the northern part of the Gulf of Bothnia and Gulf of Finland (Feistel et al., 2010). In deep areas of the Baltic Sea Proper there is a permanent halocline somewhere between 60 and 80 meters depth. Occasional saline water inflows (Major Baltic Inflows) from the North Sea (Matthäus et al., 2008) and continuous inflow of fresh water from the rivers (Ahlnäs, 1961; Hansson et al.,

2011) keep the stratification strong and cause the deep water to be anoxic in the Baltic Proper and western Gulf of Finland.

Utö Island (59º 46'50 N, 21º 22'23 E) where the observations of this study are made, is located at the outer edge of the Archipelago Sea towards the Baltic Sea (Fig. 1). Utö is the southernmost permanently inhabited island in the Finnish archipelago. Its land area is approximately 0.8 km$^2$ and the average permanent population between 20 and 30 people. Utö Island has had a lighthouse and a pilot station since 18th century, with permanent pilots living on the island for generations. Due

to its location and permanent population, local pilots and during the 20th century, military officers have carried out observations on daily basis except some short breaks during the WWI.

The sea area 1 km northwest of Utö is relatively deep, 104 m, and is connected to the open sea through a deep channel, so especially the deep samples may be considered to reflect the properties of Northern Baltic (Ahlnäs, 1961). Due to the regional bottom topography, depth structure and prevailing wind direction from southwest, currents at Utö Deep may be relatively strong

in comparison to typical values in the Baltic Sea. Biological characteristics on the area are typical for the outer archipelago, with almost annual cyanobacterial algae blooms in July or August (e.g. Kahru et al., 1994; Seppälä et al., 2007). During the last decades, sea ice is observed at Utö every few years (Jevrejeva et al., 2004).

Finnish Meteorological Institute started meteorological observations at Utö already in 1881 and fixed oceanographic station has been in operation since 1900. Atmospheric trace gas and aerosol measurements were started on the island in 1980 in the

framework of European Monitoring and Evaluation Programme (EMEP) (Ruoho-Airola et al., 2003; Laurila and Hakola, 1996). In 2012 Finnish Meteorological Institute (FMI) and Finnish Environment Institute (SYKE) started the construction of a marine research station on island, leading to a combined Utö Atmospheric and Marine Research Station (Finnish Meteorological Institute, 2017). The list of current continuous observations at Utö is given in Appendix A.

Currently, Utö station is part of the HELCOM marine monitoring network, Integrated Carbon Observing System (ICOS),

Finnish Marine Research Infrastructure (FINMARI) and Joint European Research Infrastructure network for Coastal Observatory – Novel European eXpertise for coastal observaTories (JERICO-NEXT) (Puillat et al., 2016). It is also planned to become a part of European Research Infrastructure for the observation of Aerosol, Clouds, and Trace gases (ACTRIS).

## 3   Observations and methods used in this study

### 3.1   Observations and other data

In this study, we focus on long-term changes at Utö during the period 1881-2016. As we do not have all data available for the whole period, and there are gaps in the data, best coverage for combined data is for the period 1911-2016.

Meteorological observations including air temperature, wind speed and direction, cloudiness and sea surface temperature have been carried out with developing methods since 1881, initially three times a day and later on with increasing time reso-



lution, currently with 10 minutes logging interval. As the methodology and exact observing places have changed during this long period and there is limited metadata from early measurements, especially the older data contains uncertainties. Meteorological data for the period 1881-1959 used in this study was manually digitized from annual written records; data since 1959 is taken from FMI electronic archives. These data were automatically checked from clear outliers and bad data, in addition to the

5 original quality assurance done for all FMI operational meteorological observations. Our study focuses on air temperatures, but cloudiness and winds are also shortly discussed.

Vertical profiles of sea water salinity and temperature have been measured at Utö deep (59º 46'58 N, 21º 20'58 E) at standard depths down to 100 meters. The depth of the deepest measurement has varied, but was always more than 80m. The temperature observations started in 1900 and salinities have been measured since 1911. There were also some salinity records from the

10 period 1900-1911, but due to poor data quality, they are excluded from our analysis.

The routine observations (until 2003) were done in fixed oceanographic stations in principle with ten days intervals, in every month's 1st, 11th and 21st day. Because the observations from small boat are weather dependent, the exact observation days sometimes differ from the scheduled and also the number of winter observations is smaller than that of the summer observations. Thus, the results for winter months include higher uncertainties than for the other seasons.

There was a gap in the marine observations in 2003-2012, because of lack of observers at that time-period. Observations were started again in 2013, with an RBR XR-620 CTD with a RINKO-III Dissolved Oxygen sensor, measuring temperature, salinity and pressure with 0.5 db (~ m) intervals. The profiles have been measured once in every ten days when weather and ice conditions have permitted. These new data are combined with the earlier fixed depth data in order to obtain as long time series as possible.

Oceanographic profiles were visually inspected, specifically for this study, using a code written in MATLAB. All suspicious profiles, like those with clearly wrong salinities and/or temperatures, or impossible density profiles were rejected. After the quality check, we have 1520 good-quality full vertical profiles of temperature and salinity from the period 1911-2002 and 112 more profiles from the period 2013/04 – 2017/07.

Sea ice data were obtained from ice maps done during the ice season for the Gulf of Finland. The ice charts are based on ice

observations done in large number of locations, one of the observing sites being Utö. We used this generalized ice data instead of direct local ice observations from Utö, as there were few periods when we were not able to determine whether the sea ice observations at Utö were missing, or there were simply no ice. For the same reason, we also excluded from our analysis the ice thickness observations made at Utö during the period 1897-2015. The use of ice charts also provided us with better general understanding of the ice situation in the vicinity of Utö. The ice data used in this study are based on 8564 manually-analyzed

ice maps from the period 1914 - 2016.

NAO data was taken from Jones et al. (1997) with updates from Osborn (2004, 2006, 2011). Major Baltic Inflow (MBI) data is taken from Matthäus et al. (2008), with latest updates from Mohrholz et al. (2015) and Nauman et al. (2016).



## 3.2 Time-series analysis methods

A trend is a change in the statistical properties of the background state of a system (Chandler and Scott, 2011). The simplest case is a linear trend, in which, when applicable, we need to specify only the trend coefficient and its uncertainty. Natural systems evolve continuously over time, and often, it is not appropriate to approximate the background evolution with a constant

trend. Furthermore, the time series can include multiple time dependent cycles, and they are typically non-stationary, i.e., their distributional properties change over time.

In this work, we applied dynamic linear model (DLM) approach to time series analysis of multiple meteorological variables measured at Utö Island. Dynamic linear models are linear regression models whose regression coefficients can depend on time. With a properly set-up and estimated DLM model, we can detect significant changes in the background states and estimate

the trends. The magnitude of the trend in an individual model is not prescribed by the modeling formulation. This dynamic approach is well known and documented in time series literature (Chatfield, 1989; Harvey, 1991; Hamilton, 1994; Migon et al., 2005). The method is the same that was already applied in Mikkonen et al. (2015) for the Finnish mean temperature time series. DLM is used to statistically describe the underlying processes that generate variability in the observations. The method effectively decomposes the series into basic components, such as level, trend, seasonality, and noise. The components can be

allowed to change over time, and the magnitude of this change can be modeled and estimated. The part of the variability that is not explained by the chosen model is assumed to be uncorrelated noise and we can evaluate the validity of this assumption by statistical model residual diagnostics.

The model provides a method to detect and quantify trends, but it does not directly provide explanations for the observed changes, i.e., whether for example natural variability could explain the changes in the background levels. The model construc-

tion procedure and equational formulation follows closely the one described in Mikkonen et al. (2015) and the results were calculated with software package DLM for R statistical language described in Petris et al. (2009) and Petris (2010). Confidence limits for the trend estimates were calculated with Maximum Entropy Bootstrap for Time Series method (Hrishikesh and López-de Lacalle, 2009). The variables of interest in this study were air temperature, cloudiness, seawater temperature in different depths, water salinity and density. Each variable was inspected in both manners: as total measurement series, where

monthly variation is included in the model, and separately in different seasons of the year. In aim to keep the focus of this paper solid, we focus on a selection of the variables in stead of all possible seasonal data.

## 4 Results

### 4.1 Long-term changes in atmospheric temperatures

Fig. 2 represents the annual average atmospheric temperature at Utö during the period 1881-2016, together with the mean values

for whole Finland. For illustrative purposes, we also included 5-years running mean (requiring at least 40% data coverage) in this and subsequent figures; however, quantitative results are based on DLM analyses only. According to DLM analysis, annual average temperature at Utö has increased from 6.0 °C in 1881 to 7.5 °C in 2015. The total increase would have been





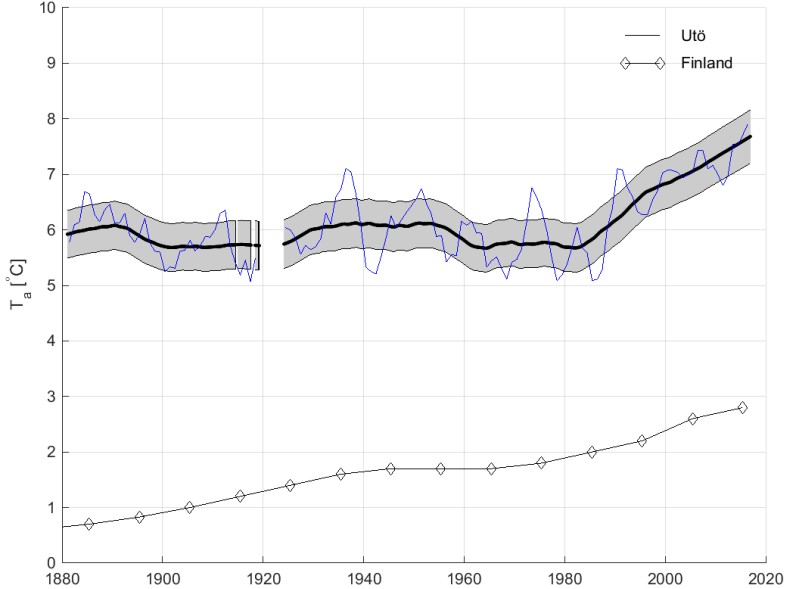

**Figure 2.** Average annual air temperatures at Utö during 1881-2016. Solid line represent temperatures calculated with DLM and the gray area shows the 95 percent confidence range calculated using bootstrap method. Blue line over the DLM curve is 5-years running mean of atmospheric temperature at Utö. Thin black line with diamonds shows the decadal average temperatures calculated for whole Finland (Mikkonen et al., 2015).

0.11 °C/decade if it were linear, which is lower than the average increase 0.14 °C/decade observed in Finland (Mikkonen et al., 2015).

While in Mikkonen et al. (2015) the temperature increase follows the pattern in global temperature time series (NASA, 2017), where the warming has taken place in two periods, before 1940's and after 1960's, in Utö the temperature increase has taken place only since 1980 without observable trend before that. This leads to an increase of 0.4 °C/decade during the last 35 years period, in line with results reported by Lehmann et al. (2011) and Almén et al. (2017), and the concluding remark of Mikkonen et al. (2015) stating that within the last 40 years the rate of temperature change in Finland has varied between 0.2 and 0.4 °C/decade.

We also investigated the annual average temperatures against the NAO indices (Fig. 3) (Hänninen et al., 2000; Osborn, 2004, 2006, 2011), and found for individual years, as expected (Lehmann et al., 2011), lower temperatures connected to highest negative NAO values and vice-versa (visible also in 5-years running mean shown in Figs. 2 and 3). However, we were not able to explain the temperature trend or the longer (> 10 years) periods with higher and lower temperature with the NAO cycle.





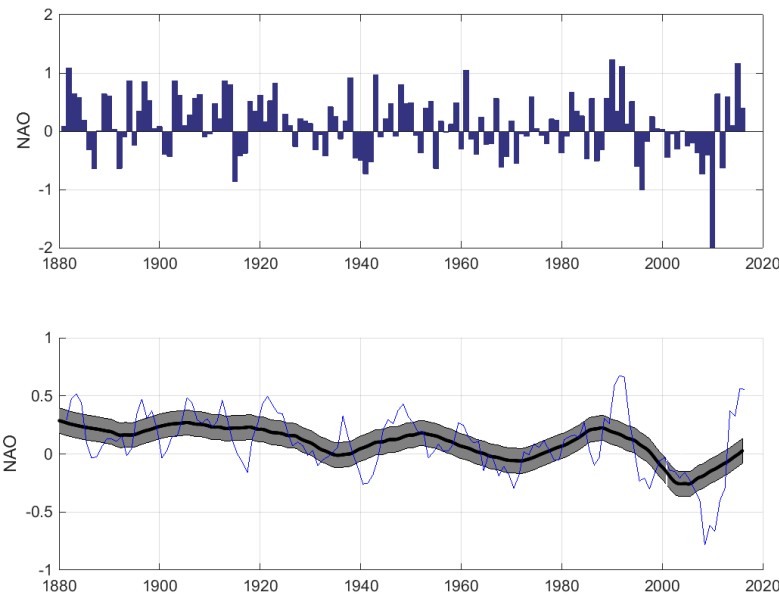

**Figure 3.** North-Atlantic Oscillation (NAO) during 1880-2016. Upper panel shows yearly NAO values and lower panel shows the trend of the NAO, together with 5-year running average values.

In addition to overall temperature trend, it is of interest to look the changes in different seasons (Fig. 4). Using simply three calendar months as seasons, we see similar trends in each season as in the annual temperature (Fig. 2). Looking on individual seasons, we notice that the long-term increase in annual temperatures (Fig. 2) results especially from the increase of temperatures during the winter and spring.

## 4.2 Cloudiness and wind

The quality assurance for atmospheric temperatures is relatively easy. For cloudiness and especially wind, the situation is more complicated. We analyzed the changes in cloudiness for the period 1881-2005 for which we had manual observations available. After October 2005 the cloud observations have been done with a ceilometer, and the results are not comparable with the previous data. Fig. 5 shows the time series of cloudiness on the scale from 0 to 8. We see increase till 1990 and after that a decrease until the end of our manual observations, 2005. Automated observations (not shown) since 2005 show again an increasing trend in cloudiness. Further investigations focusing on reasons behind the changes in cloudiness are, however, out of scope of this paper.

We also looked at wind speed time series and found no significant changes, in line with a recent study for the period 1979–2008 (Laapas and Venäläinen, 2017). However, because the wind observations are very sensitive to inhomogeneities in




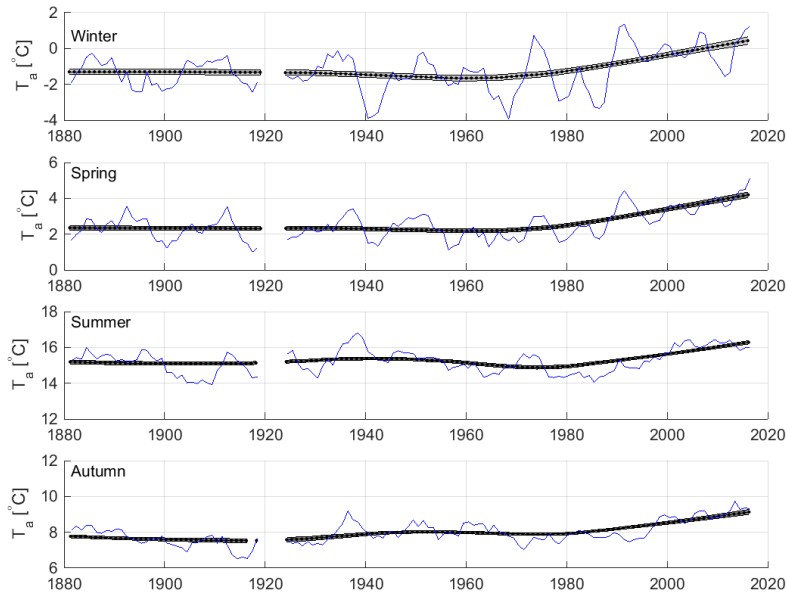

**Figure 4.** Average seasonal air temperatures at Utö during 1881-2016. Winter: Dec, Jan, Feb; Spring: March, April, May; Summer: June, July, August; Autumn: September, October, November. As in Fig. 2, Black line represent DLM-calculated trend, gray area error for the trend and blue line 5-years running average.

methods and location (Pryor et al., 2009; Wan et al., 2010; Feser et al., 2015; Laapas and Venäläinen, 2017), more analyses for observations done prior to 1959 are needed before further use of this part of the dataset.

### 4.3 Long-term changes in sea water temperatures, salinities and sea ice

Fig. 6 shows the monthly median sea water temperatures, salinities and densities in Utö deep during the period 1911-2016.
5  From the figure, we see the annual cycle of the water body: strong vertical stratification in the summer, with mixed layer depths around 20 meters; vertical mixing throughout the whole water body in October; and the seasonal variation of salinity in all depths. Generally, at our site, average sea surface temperature varies between 18°C during the summer and 0 °C during the winter while bottom temperature range is from 2 °C to 5 °C. Surface salinities vary between 6‰ and 7‰ , water being less saline during the summer and more saline in winter. At the bottom, situation is opposite, with up to 8 ‰ in summer and
10  around 7‰ in winter. Density follows the cycle of salinity in deep water. The values for individual years and months may be significantly different from these medians due to Major Baltic Inflows (MBI) bringing large amounts of saline water to the Baltic Sea, variation of annual temperatures, river inflow, and existence of sea ice cover. In contrast to the Baltic Sea Proper, we do not see a permanent halocline between 60 and 80 meters depth (Leppäranta and Myrberg, 2009). This may result from the relatively strong currents at Utö deep.





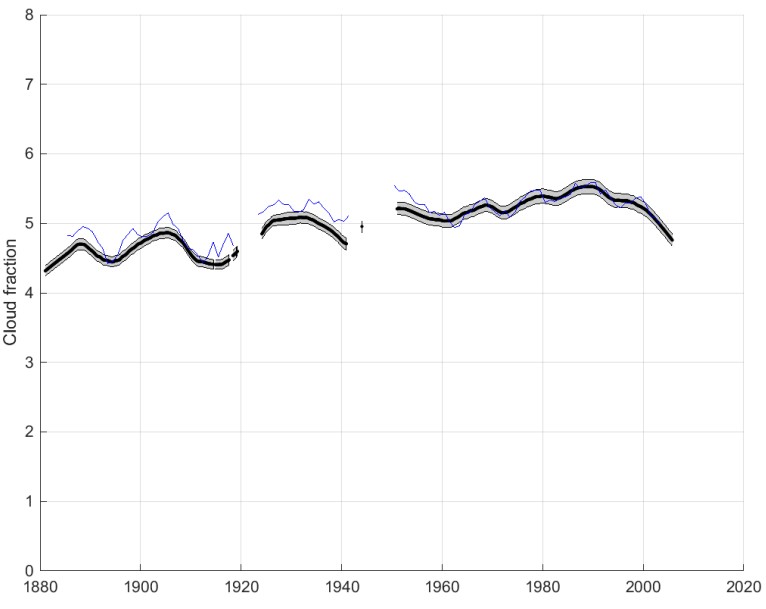

**Figure 5.** Cloud fraction during 1881 - 2005

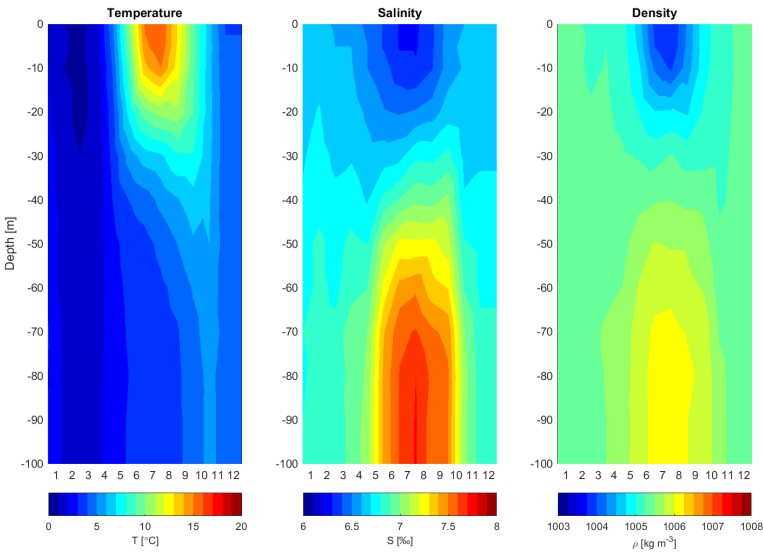

**Figure 6.** Monthly median sea water temperature, salinity and density in Utö during 1911-2016.



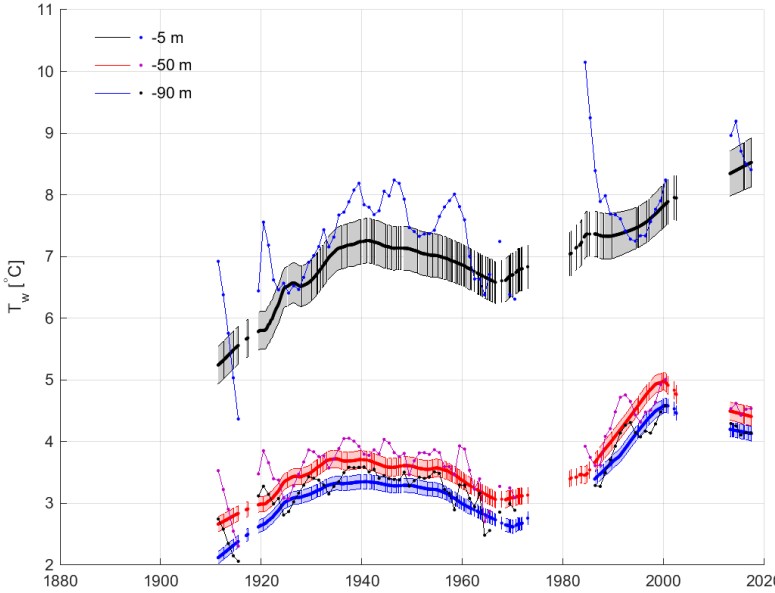

**Figure 7.** Average sea water temperatures in Utö at depths of -5, -50 and -90m. For better visualization, we have used a combination of black line and blue for -5 meters, red and magenta for -50 meters and blue and black for -90 meters as shown in the legend.

Exact measurement depths and the number of different observation depths have varied during the last one hundred year. As our focus is on long-term trends, we decided to focus on three different depths where we have most data, while the depths have also physical meaning: 5 m represents the sea surface layer which is quite directly influenced by the atmosphere, but is most probable not influenced by measurement errors, 50 m depth which is at the old winter water layer that is not directly influenced

by the surface processes in summer and that is also the middle point between the surface and bottom, and 90 m which is the closest point to the bottom, with high data coverage. While the decision to investigate 5m and 90 m depths was clear, selection of 50 m was also supported by manual analysis of mixed depths: we calculated the top and bottom of mixed layer depths using similar method that Lips et al. (2016) utilized, and found that during the summer period (June, July and August) the bottom of thermocline was always above 50 meters depth.

Fig. 7 represents the trends in water temperatures at these three depths. We see that the surface temperature follows the behavior of atmospheric temperatures (Fig. 2), with a rapid increase since 1980's and a warmer period from 1930's until 1960's. The overall increase has been approximately 0.3 °C/decade during the last 100 years. For deeper layers, we observe partly different trends, with a faster increase in temperatures in 1980's and a drop or hiatus during the last few years. As the main heating to the sea water comes from the surface of the sea, the higher increase of deep water temperatures during

the 1980's and 1990's, and recent decrease have to be influence by other phenomena than simply the increasing atmospheric temperatures.

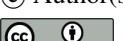


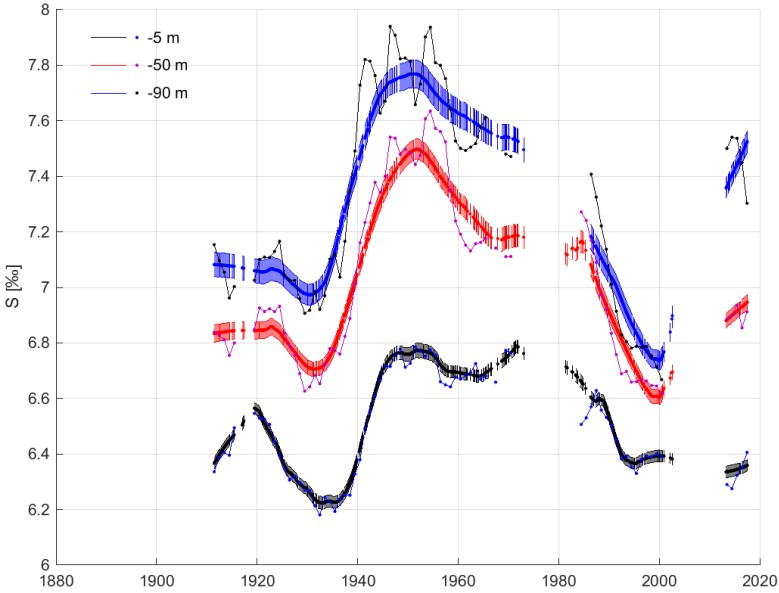

**Figure 8.** Average sea water salinities in Utö at -5, -50 and -90 m depths. For details, see Fig. 7

Fig. 8 shows the changes in salinity at different depths. We see that the salinity has varied significantly during the observing period but there is no general trend in the data. However, there are clear changes in the salinity stratification: the stratification was largest in 1950's and smallest in 1980's and 1990's during the stagnation period when no major Baltic inflows occurred. The stratification increased again since 2013.

5     The sea water temperatures and salinities combine in the Fig. 9 as sea water density. As the salinity is a key factor (in layers where temperature variations are small) influencing sea water density, the density curves follow the changes in salinity. Fig. 9 shows smaller vertical density gradients in the 1980's and 1990's than earlier, and increase of density stratification during the last few years.

    These changes, we assume, are responsible for the increased water temperatures at 50 m and 90 m depths during the period 10  1980-2000 and the recent decrease since 2012, and an explanation why there is a difference between the behavior (slope) of surface water being directly in contact with the atmosphere and the deeper water layers during this period, 1980-2000.

    The observations at Utö are insufficient to directly explain the reasons for changes in the salinity stratification. However, the frequency and strength of Major Baltic Inflows (Feistel et al., 2008, update based on Nauman et al., 2016), shown in Fig. 10, clearly explain that the changes are related to major Baltic Inflows since 1980's. We also see a rapid increase in 1940's. While 15  there is no MBI data due to WWII (Matthäus et al., 2008), study by Ahlnäs (1961) supports strong MBI's during that period, combined with reduced river discharge (Hansson et al., 2011).





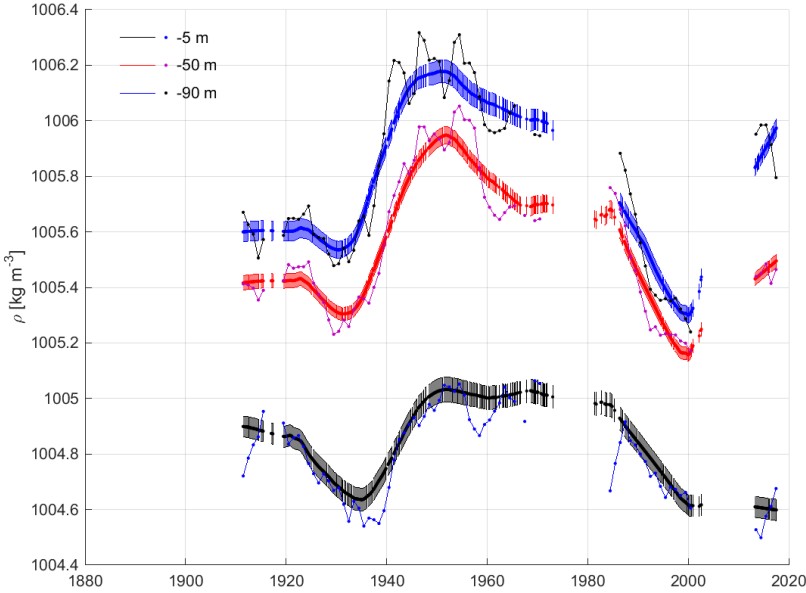

**Figure 9.** Average sea water densities in Utö at -5, -50 and -90 m depths. For details, see Fig. 7

Utö is at the border between open Baltic Sea proper and the Archipelago Sea. The sea ice cover (Fig. 11) has thus varied a lot from year to year. We see that the average duration of sea ice cover has decreased with 50% since 1980, in line with the increased average temperature (Fig. 2) and previous studies (Jevrejeva et al., 2004; Merkouriadi and Leppäranta, 2014). This decrease in ice cover may have enhanced the recent rapid increase in air temperatures at Utö since open sea is a large source of

5 latent heat and reduces the lowest temperatures observed in winter (Fig. 4). Ice cover also increases albedo, which may have influenced the surface temperatures during the spring.

Finally, we calculated average monthly air and sea surface temperatures at Utö for four different 30-years reference periods, 1891-1920, 1921-1950, 1951-1980 and 1981-2010 (Fig. 12 and Table 1). The averages show the recent warming of winters and springs with a small increase in springtime sea water temperature. In time, the date when average air temperature exceeds

5 °C has changed to 5.4 days earlier from the previous periods to 1981-2010. The date when seawater temperature at 5 m depth exceed 4 °C has changed even to 8.8 days earlier from the period 1951-1980 (and 1921-1950) to 1981-2010.

Average air temperatures, cloud fractions, sea water temperatures and salinities and durations of sea ice cover, together with standard deviations for the different 30-years periods are given in Table 1. The data show clearly the high natural variability of environmental variables. However, these averages also hide the rapid change which has taken place especially in temperatures

since 1980, which was clearly visible in figures showing the trends.



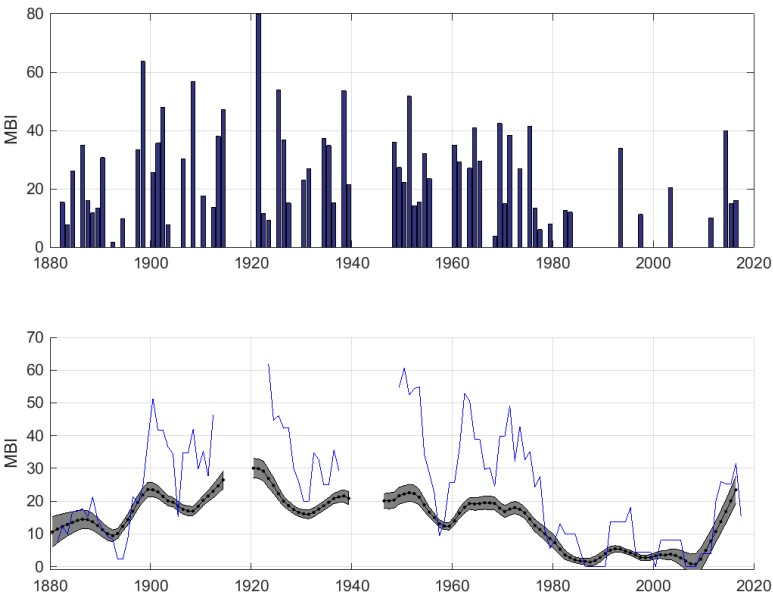

**Figure 10.** Major Baltic Inflows (MBI) during 1880-2016. There is no data available for the World War periods 1915-1920 (WWI) and 1940-1946 (WWII).

**Table 1.** Average values with standard deviations for the 30-years periods 1891-1920, 1921-1950, 1951-1980 and 1981-2010. As there have been gaps in observations, the uncertainties between variables and periods vary. Due to limited amount hydrographic and sea ice data during the first period (1891-1920), only values for air temperature and cloud fraction are shown.

| | | 1891-1920 | 1921-1950 | 1951-1980 | 1981-2010 |
|---|---|---|---|---|---|
| Air temperature (°C) | | 5.77 (7.56) | 6.15 (7.91) | 5.81 (7.64) | 6.56 (7.33) |
| Cloud fraction (0...8) | | 4.73 (5.75) | 5.20 (3.25) | 5.24 (3.04) | 5.33 (2.82) |
| Sea water temperature (°C) | -5 m | - | 7.31 (5.47) | 7.16 (5.47) | 8.11 (5.59) |
| | -50 m | - | 3.70 (2.08) | 3.52 (1.94) | 4.44 (2.56) |
| | -90 m | - | 3.33 (1.63) | 3.13 (1.48) | 4.07 (2.11) |
| Sea water salinity (‰) | -5 m | - | 6.43 (0.34) | 6.72 (0.28) | 6.45 (0.29) |
| | -50 m | - | 6.99 (0.48) | 7.36 (0.48) | 6.83 (0.35) |
| | -90 m | - | 7.31 (0.69) | 7.67 (0.63) | 6.91 (0.36) |
| Sea ice cover duration | Days | - | 31.8 (36.5) | 37.5 (34.7) | 27.8 (33.0) |

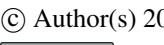



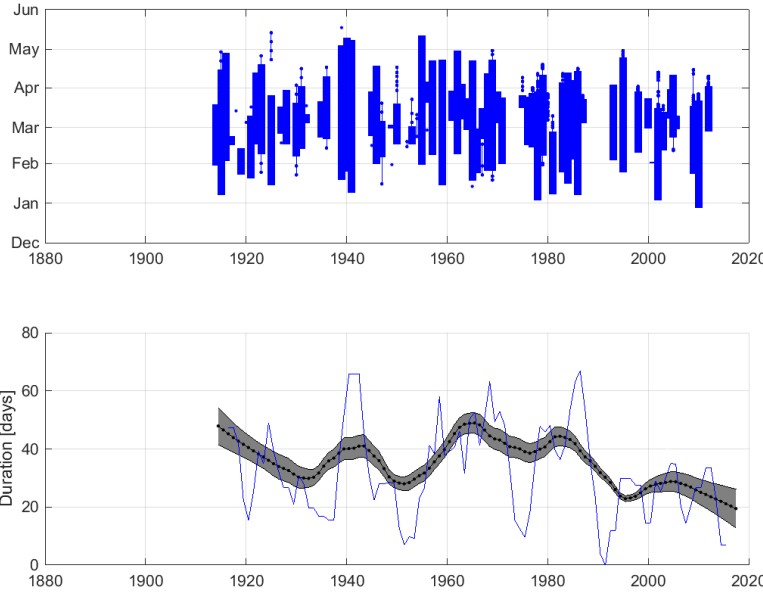

**Figure 11.** Winter time sea ice cover at Utö. The upper panel shows the yearly duration of the ice cover and the lower panel shows the DLM trend and five year moving averages of the length of the ice season.

## 5    Conclusions

In this study, we used about 100 years long time series of atmospheric and marine observations carried out at Utö. The focus was on long-term changes and potential impacts of warming climate to the Baltic Sea hydrography. In an earlier study by Mikkonen et al. (2015), a clear increase of atmospheric temperatures was observed throughout continental Finland. On the sea areas, however, changes are dampened by the large heat capacity of the sea. In winter, the sea ice influence albedo, and sensible and latent heat fluxes between the sea and the atmosphere.

In our study, we saw an increase in the atmospheric and sea (surface) water temperatures only since 1980's, which is different from the Finnish average air temperature increase observed throughout the 20th century. As the increase observed at Utö is mostly due to the warmer spring and winter months, we assume that the impact of warming climate is visible especially after the reduction of winter time sea ice cover.

We also found that there was a clear reduction in the vertical stability of the water column during the so-called stagnation period 1980-2010, when there were less Major Baltic Inflows than before. This enhanced mixing, together with increased air temperature may have been responsible for the increased deep water temperatures during this period. Latest observations since 2013 show again an increase in vertical stratification due to recent MBI's, which have increased the bottom salinities and decreased the temperatures due to reduced vertical mixing.



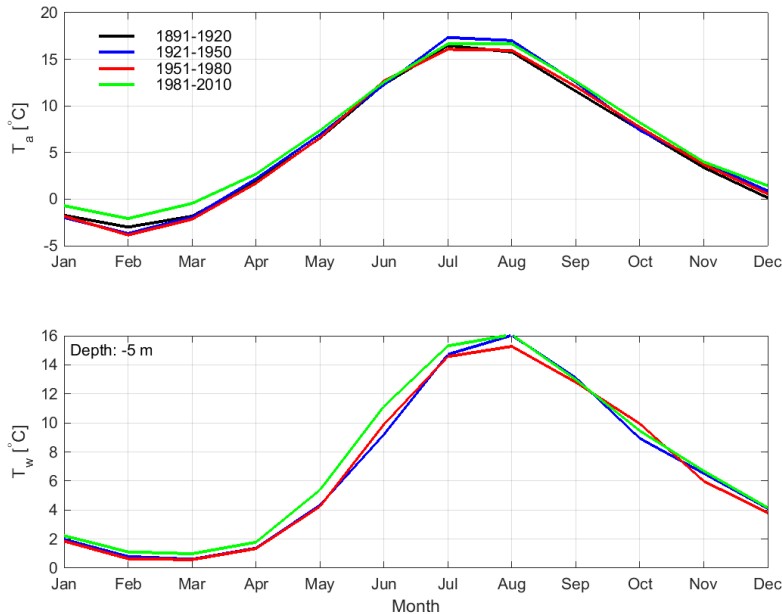

**Figure 12.** Monthly average air temperatures (upper panel) and sea surface water temperatures (lower panel) at Utö during the four 30-years reference periods

Our results are in line with previous studies on climate and hydrographic changes on the Northern Baltic Sea region. The data and analysis represented in this study also forms a solid base for detailed process studies which are an integral part of the JERICO-NEXT concept of integrated coastal observatories (Puillat et al., 2016). In the future, our aim is to continue the analyses with other methods, and studies focusing more on individual changes and variability in frequency domain using wavelet methods.

*Data availability.* The data used in this article will be available through Finnish Meteorological Institute open data portal (https://en.ilmatieteenlaitos.fi/open-data) after the publication of this article.

**Appendix A: Observations at Utö Atmospheric and Marine Research Station**

*Acknowledgements.* First of all, the authors want to thank the five generations of observers, pilots and soldiers of keeping the observations running. Especially we thank our recent observers Ismo and Brita Willström who have done and maintained the measurements used in this





**Table A1.** Continuous atmospheric and marine observation at Utö. References for some earlier publications utilizing observations from Utö are given in the last column.

| | Variable | Start (year) | Reference |
|---|---|---|---|
| Meteorological observations | T, p, WS, WD, RH | 1881 | (Laapas and Venäläinen, 2017) |
| | Precipitation, cloudiness | 1881 | |
| | Global radiation | 1998 | (Riihelä et al., 2015) |
| | Diffuse radiation | 1998 | |
| | UV-radiation | 1998 | |
| | Visibility | 2002 | |
| | Cloud cover and height | 2006 | |
| | 3-D wind profile (Doppler lidar) | 2012 | (Hirsikko et al., 2014; Tuononen et al., 2017) |
| | Weather camera | 2014 | |
| Aerosol and trace gas observations | Aerosol mass ($PM_{10}$) | 1980 | (Ruoho-Airola et al., 2003) |
| | $SO_2$ | 1980 | (Ruoho-Airola and Salmi, 2001) |
| | Aerosol chemical composition ($PM_{10}$) | 1980 | (Ruoho-Airola et al., 2003) |
| | $NO_x$, $O_3$ | 1986 | (Engler et al., 2007) |
| | Aerosol mass ($PM_{2.5}$) | 2003 | |
| | Aerosol size distribution | 2004 | (Engler et al., 2007; Hyvärinen et al., 2008) |
| | Aerosol absorption | 2007 | (Hyvärinen et al., 2011) |
| | Aerosol scattering | 2010 | (Hyvärinen et al., 2011) |
| | Aerosol chemical composition ($PM_{2.5}$) | 2011 | |
| | Phosphorus deposition | 2014 | (Makkonen et al., 2015) |
| | Radon | 2015 | (Vesterbacka, 2017) |
| Atmospheric greenhouse gas measurements | $CO_2$, $CH_2$ and CO-concentrations | 2012 | (Kilkki et al., 2015) |
| | $CO_2$- flux | 2012 | (Honkanen et al., 2018) |
| Marine observations | Sea ice observations | 1897 | (Seinä and Palosuo, 1996) |
| | Temperature and salinity profiling 0-90m | 1900 | (Ahlnäs, 1961) |
| | Nutrient and chlorophyll profiles 0-70m | 2001 | (Suomela, 2003) |
| | Sea Ice radar | 2013 | |
| | Temperature, salinity, $O_2$, turbidity, chlorophyll (from 5 m depth) | 2014 | |
| | Currents (0...-23m) and surface waves | 2014 | (Haavisto, 2015) |
| | Automatic Identification System (AIS) | 2015 | |
| | Bottle sampler | 2015 | |
| | Spectrometric observations of phytoplankton properties | 2016 | |
| | $pCO_2$ | 2016 | (Honkanen et al., 2018) |
| | pH | 2016 | |
| | DIC | 2016 | |




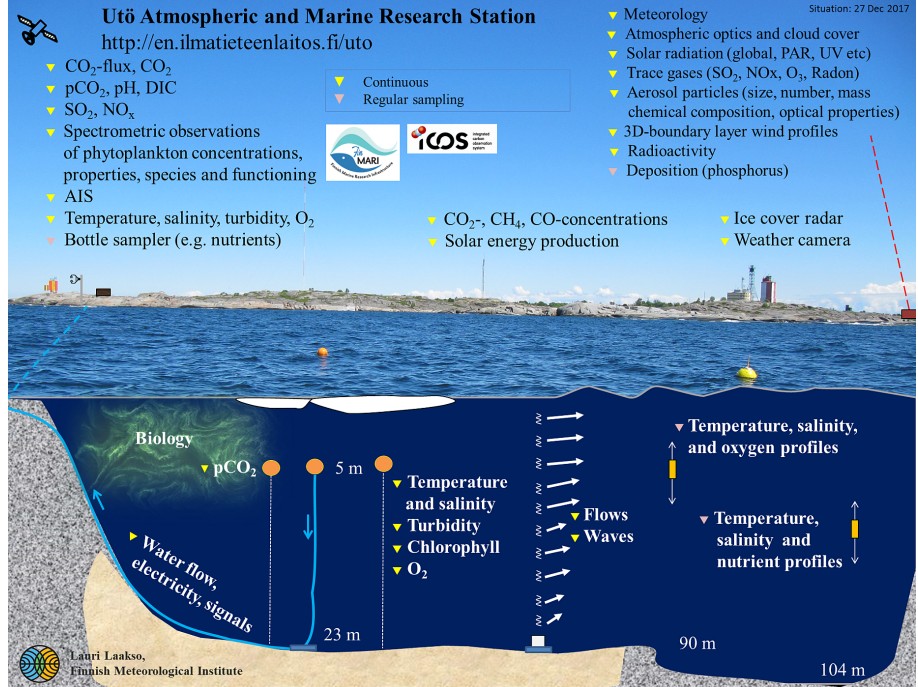

**Figure A1.** A schematic picture showing the observations at Utö and listed in Table A1

study. It is a privilege to work with this unique dataset obtained by their invaluable efforts. This project was partially funded by H2020-project Jerico-Next (grant agreement No 654410), Bonus-Integral (funded by BONUS Blue Baltic). Santtu Mikkonen acknowledges funding from Nessling Foundation and Finnish Academy (project No. 307331). MSc Sakari Äärilä is acknowledged for drawing the map of Figure 1.





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
