# Peer review of "One hundred years of atmospheric and marine observations at Finnish Utö Island in the Baltic Sea"

_Ocean Science, 2017_

## Referee Comment (RC1) · Anonymous Referee #1 · 26 Feb 2018

Laakso et al. One hundred years of atmospheric and marine observations at Utö Island, the Baltic Sea Ocean Sci. Discuss., https://doi.org/10.5194/os-2017-105

Review

The paper presents long-term meteorological and oceanographic data from the Utö station in the north-eastern part of the Baltic Proper (Archipelago Sea). It is a very valuable source of information for meteorologists, oceanographers, and climate researchers. I strongly recommend publication of the results.

However, the paper could be improved by adding more scientific discussion points. My main concern is that the statistical analysis results are presented, but not properly discussed. Much more can be done/analyzed that would help to put the results into a
wider context.

Physical processes, which could cause the observed changes in stratification and deep layer characteristics are oversimplified in the analysis – mostly the changes are explained by vertical mixing. As shown in the Gulf of Finland, the stratification very much depends on wind conditions – winds from a certain direction tend to strengthen the stratification and opposite direction weaken it. Thus, bi-directional lateral transport is an important factor. I think, a look at the topography of the study area (and connections, sills between the Utö Deep and Baltic Proper) could be relevant.

Also, as mentioned by the authors, this analysis sets the background for the further studies of biogeochemical changes. But no suggestions are made on this subject in the paper/discussion.

Specific comments

Abstract

There are some spelling errors in Abstract. Please, correct them.

P1, L14-15: The last sentence of Abstract has to be rephrased. If I understand the results correctly, the ice does not cause large local effects anymore in this new phase. In the present form, the sentence has an opposite meaning that the ice does not reduce local effects. Did it reduce or cause the local effects earlier?

1. Introduction

I miss a broader problem setting. Also, a scientific aim of the study could be formulated.

2. Measurement site and general characteristics

P2, L29: I would avoid using the term "seasonal sea" which is not commonly used in the scientific literature.

P3, Fig. 1: It is quite empty. At least the sub-basins of the Baltic mentioned in the

text should be shown (Archipelago Sea, Baltic Proper, Gulf of Finland, etc.). Also, consider presenting a local map where the oceanographic measurement site with the topography of the surrounding area could be seen.

P4, L3-5: I agree that saline water inflows and freshwater input keep the stratification strong, but I do not agree that it causes the deep water to be anoxic. It could be opposite – saline water inflows could ventilate the deep layer. Moreover, the main reason for oxygen depletion is consumption of oxygen.

P4, L13-15: It is not obvious how the bottom topography and prevailing winds cause strong currents in the Utö Deep. Please, give a reference or explain it (also providing a map with topography if appropriate).

3. Observations and methods

P4, L28-29: Please consider other options for sub-titles, e.g., "Observations and methods" instead of "Observations and methods used in this study"

P6, L25-26: I do not understand the last sentence of this sub-chapter "In aim to keep the focus of this paper solid, we focus on a selection of the variables in stead of all possible seasonal data." Please, rephrase it.

4. Results

P7, L9-13: Is the annual mean NAO index the best parameter to use here? For instance, Lehmann et al. (2011) did use NAO winter index (from December to March). Also, I do not see that the highest negative NAO values and low temperatures are connected (e.g., 2009 has far lowest NAO but not the lowest temperature).

P9, L9 and Fig. 6: Why median values are used here?

P9, L13-14: I do not agree with the suggestion that the relatively strong currents are the reason for the absence of the halocline. Could the bottom topography restrict saltier water transport into the Utö Deep?

P11, L7-9: What do you mean by "we calculated the top and bottom of mixed layer depths"? I did not find a method in Lips et al. (2016) for that.

P11, L13-16: What could be these other phenomena responsible for deep water temperature increase in the 1980s and 1990s and recent decrease?

P12, L3: Stratification should not be large and small, but rather strong and weak.

P12, L9-11: Do you explain the observed changes in deep water temperature by vertical mixing and stratification only, or is it possible that these changes are related to lateral exchange? Please, explain it because it is not clear what are "these changes ... responsible for the increased water temperatures at 50 m and 90 m depths".

P12, L14: What is meant by "We also see a rapid increase in 1940s"?

P13, L7-8: Why these 30-year periods were selected for the comparison. It could be reasonable for the atmospheric data, but not for the oceanographic parameters which revealed a rapid change in the 1990s.

P13, L14-15: Has this sentence ("However, ...") the meaning of the previous comment that the chosen periods hide the rapid change after the 1980s?

P14, Table 1: How these averages and standard deviations were estimated?

5. Conclusions

P15, L15: The saline water inflows increase salinity in the deep layer, but they could cause either an increase (mostly) or decrease in temperature. This change is not directly connected to the reduced vertical mixing. Do you have any pieces of evidence that the observed decrease in deep water temperature was due to the reduced vertical mixing?

---

## Referee Comment (RC2) · I. Vuorinen (Referee) · 3 May 2018

General comments:

The paper presents first time digitized and quality assured oceanographic data from the Northern Baltic proper in (semi) open sea conditions. Temporal range of the data is impressive, starting in 1881. Spatially the data is from one spot, which lies somewhat mid-way between open sea and coastal conditions, also between the Bay of Bothnia and the Gulf of Finland. Approach and methods are basic, which is okay for this type of paper (presentation of a new, significant data set). The discussion could be somewhat more extensive (see below in specific comments).

Presentation of figures and tables is appropriate (with some exceptions which are commented below), and English is mostly okay. I suggest below some places where wording should be reconsidered.

I suggest publication with minor corrections.

Specific comments

Title: One hundred years of atmospheric and marine observations at Utö Island, the Baltic Sea. -There are several islands with that name in the Baltic. I know of one in Sweden and two in Finland. Consider adding the coordinates, the country, or other information in the title in order to avoid mixing at least with the Utö in the Stockholm archipelago.

Abstract: I like the last sentence. It points out a possible tipping into a new phase. This idea should be discussed more thoroughly, considering a possible breakpoint, its temporal location and affecting mechanisms. I agree, that would be obligatorily speculative, but as at present this idea seems to be the only one outcome suggested by authors, it would be important to ponder it more deeply. I miss other conclusive sentences, such as what would be the best, or more appropriate way (instead of just assuming a linear model) to analyze the evidently non-linear change over time which is seen in many parameters, such as in the salinity. I agree that the linear analysis should be the one to start with, but I also expect the authors to show capability to go further. Seeing abrupt changes like temperature since the 1980s and salinity at Utö makes me look for possible explanations and coincidences. You could suggest a way forward, and the use of e.g. breakpoint equations in coming analyses, with other, non-linear, models.

Page 2 lines 10-12, you aim the paper to "analyse these time series in order to get information on typical atmospheric and marine conditions", but reading the paper makes me think that several less typical phenomena are shown, such as a rapid increase of salinity, or a decrease and disappearance of the ice cover and a subsequent suggestion of a shift of balance in the climate of Utö into a new phase. So I suggest rewording these

lines, for the reader not expect too much of "just typical" happenings being observed.

Line 31, you give the coordinates and write about the observation site and about the Island. Are these coordinates for the midpoint of the Island or the lighthouse or coordinates for atmospheric observations? Compare to page 4, line 33, where you give coordinates for the oceanographic sampling point.

Page 3, the map should have two panels, one showing the location in the Baltic sea (the present one) and another to show local bottom topography, depth etc.

Page 4, line 1, "with permanent pilots living on the island for generations" this is repetition of the information of the first part of the sentence, and, besides, "pilots living for generations" sounds improbable. Remove the sentence.

Line 5, you write that: especially the deep samples may be considered to represent conditions of the Baltic Proper. On the other hand you write (page 9, line 1-2): we do not see any permanent halocline (and comment that possible cause to the halocline missing could be mixing due to currents.) These two statements are contradictory, first one is by Ahlnäs in 1961. Have the circumstances changed between then and now? Lack of the deep halocline also puts the sampling station oceanographically more to the Bothnian Bay side than on the Northern Baltic Sea. Could you comment on that? Line 7. I do not accept the phrase about biological characteristics. First: there is no information included in writing that "biological characteristics are typical for the outer archipelago" as this is anyway the basic assumption. Secondly, this kind of basic assumption is not valid for this location as biological characteristics point out to a eutrophic environment. Since the 1980′s the cyanobacterial blooms have been observed in this area, but before that the area, as the whole Northern Baltic Proper was considered to be an oligotrophic environment. Same rapid change from oligotrophy to eutrophy is seen in, for example, in Secchi disc readings in the Gulf of Finland during the 1900. Please give appropriate information on biological change over at least the last decades, as you do for the sea ice in the next sentence.
**OSD**

Page 6, lines 32-33, you write that you investigated the annual average temperatures against the NAO indices in Fig 3., but that figure only shows the NAO history, while temperatures are given in the Fig. 2. You also claim finding, for individual years, lower temperatures connected to highest negative NAO values, but in Fig. 2 there are no temperature values given for individual years at all. You refer also to Fig. 2 having lowest temperature values (5 year periods) in around 1980, while this period (1980) in the NAO figures just show mid values of the index. What are "highest negative NAO values" anyway? Are they just lowest values of the index, or something else? Rewrite this part. Page 7, line 9 and 12, you write about manual observations, you should write about visual observations.

Page 8, line 3, you mention not to have found significant changes in wind speed. Okay, but my personal observation from Utö station when comparing wind observations before and after the 1970s was that there was a substantial reduction of completely calm days (see attached figure which is based in Finnish Meteorological Institute observations at the Island of Utö)). So the overall windiness has increased anyway. As you suggest, more analyses are needed. You could try and include also the data on calm days.

Page 9, line 4-5, remove the sentence: As our focus is.." and start directly from: We decided to... Lines 14-15 you write that: "the surface temperature follows the….atmospheric temperatures Fig. 2) …with a rapid increase in 1980s, which is okay and correct, but then you write that: "and a warmer period from 1930 until 1960s" which, however is not seen at all in the Fig 2 which you refer to. Rewrite that part.

Page 13, line 10, "reduces the lowest temperatures" sounds strange. Consider rewording.

Typos: page 9, line 4 reads: one hundred year, should read one hundred years.

Please also note the supplement to this comment:

https://www.ocean-sci-discuss.net/os-2017-105/os-2017-105-RC2-supplement.zip

---

## Author Comment (AC1) · 25 May 2018

We thank the Referee 1 for his/her comments, which improved the manuscript. Detailed answers, together with the revised manuscript, are given in the supplement.

Please also note the supplement to this comment: https://www.ocean-sci-discuss.net/os-2017-105/os-2017-105-AC1-supplement.zip

---

## Author Comment (AC2) · 25 May 2018

We thank the Reviewer 2 for his comments, which improved the manuscript. Detailed answers, together with the revised manuscript, are given in the supplement. Please see also the detailed replies to Reviewer 1, which include some additional details.

Please also note the supplement to this comment:
https://www.ocean-sci-discuss.net/os-2017-105/os-2017-105-AC2-supplement.zip

---

## Author Response (AR1)

**Reply to referees**                                                                          Ver 19.6.2018

**Anonymous Referee #1**

Laakso et al. One hundred years of atmospheric and marine observations at Utö Island, the Baltic Sea Ocean Sci. Discuss., https://doi.org/10.5194/os-2017-105

**Review**

RC1_1: The paper presents long-term meteorological and oceanographic data from the Utö station in the north-eastern part of the Baltic Proper (Archipelago Sea). It is a very valuable source of information for meteorologists, oceanographers, and climate researchers. I strongly recommend publication of the results.

The authors thank the anonymous referee 1 for his/her positive words on this manuscript.

RC1_2: However, the paper could be improved by adding more scientific discussion points. My main concern is that the statistical analysis results are presented, but not properly discussed. Much more can be done/analyzed that would help to put the results into a wider context.

We have added more discussion and compared the results with previous studies. All addition in the manuscript are written in red font.

RC1_3: Physical processes, which could cause the observed changes in stratification and deep layer characteristics are oversimplified in the analysis – mostly the changes are explained by vertical mixing. As shown in the Gulf of Finland, the stratification very much depends on wind conditions – winds from a certain direction tend to strengthen the stratification and opposite direction weaken it. Thus, bidirectional lateral transport is an important factor.

We added more discussion and references related to this topic.

RC1_4: I think, a look at the topography of the study area (and connections, sills between the Utö Deep and Baltic Proper) could be relevant.

This is a good an a relevant  comment, which we further analyzed the following way:

1) We compared the available data between Utö and observation point LL15 ($59°10.99'$N;$021°44.80'$E, approximately 70 km SSE of Utö, max depth 130 m). Unfortunately, we do not have very well overlapping data period, but we made a short gif-animation (Figure R1) which clearly shows that the salinities start to deviate at 60 meters, indicating indeed that the water below this depth do not always represent the central Gulf of Finland.

[Figure]

Figure R1: Time series of salinity at different depths in Utö and observing point LL15. This figure is also given as a separate file.

2) However, this is not always the case: we see sometimes saline bottom water reaching our measurement location and a halocline between 50 and 70 meters depth (Figure R2). Such periods may last for several months. These situations and reasons behind are interesting, and worth of another study.

[Figure]

Figure R2: winter months (Oct - March) salinity difference between 50m and other depths (salinity at depth X divided with salinity at 50 meters).

We see that the saline water is sometimes able to reach Utö deep and there is a halocline between 60 and 70 m. However, it is observed only for short periods. This indicates in some cases, e.g. strong enough major Baltic Inflow or e.g. special situation with underwater currents, we have a halocline also at Utö deep.

Based on dataset used in this paper and the combined profiling buoy and ADCP-measurements started in April 2018, we may be able to understand this phenomenon better in the future.

3)  The sea areas surrounding Utö are military areas and the depth maps publicly available do not necessarily represent the actual depths of the sea. Based on public data, we see (new Fig. A2 in the manuscript) that there is an area with depth of ~60 m on the western side of Utö and one further south, which may prevent the deep, saline water from entering the location of hydrographic observations. We composed a map based on public data, which is now attached in the appendix.

4)  During the spring 2018, we have made some CTD-soundings (Figure R3) next to our new profiling buoy (Fig A2, location 6) and compared those with the observations on our long-term hydrographic observing location (Fig A2, location 1). The salinity profiles overlap  between the surface and 60 m depth, while the salinity between 60 and 75 m is higher in the location South of Utö

[Figure]

Figure R3: CTD profiles taken in the traditional measurement location (Figure A2, location 1: curves 0...94 m) and next to the profiling buoy (Figure A2, location 6, curves 0...75 m) in 16 May 2018.

This question is now shortly discussed in the paper

RC1_5: Also, as mentioned by the authors, this analysis sets the background for the further studies of biogeochemical changes. But no suggestions are made on this subject in the paper/discussion.

We added the following paragraph in the conclusions:

"An interesting study utilizing the time series presented in this paper together with the new observations will be to use the new cabled bottom profiler together with an ADCP (Fig. A2 and Table A1) to study the occasional inflows (briefly discussed in this paper) of saline bottom water which may have significant impacts on the Archipelago Sea ecosystem (Vuorinen et al., 2015).
        Another planned study combining hydrographic observations with biogeochemistry and climate change is to use the profiler together with the flow-through system to analyze the thickness of biologically active layer and its connection to the marine carbon cycle.

Together with our new observations, the long data series represented in this paper will support better understanding of both the earlier observations and current, on-going physical, chemical and biological changes of the Baltic Sea."

Specific comments

RC1_6: Abstract
There are some spelling errors in Abstract. Please, correct them.

Corrected.

RC1_7: P1, L14-15: The last sentence of Abstract has to be rephrased. If I understand the results correctly, the ice does not cause large local effects anymore in this new phase. In the present form, the sentence has an opposite meaning that the ice does not reduce local effects. Did it reduce or cause the local effects earlier?

Clarified. The aim is to say that we see increase in local atmospheric temperatures at Utö only after the duration of ice cover has significantly decreased.

RC1_8: 1. Introduction
I miss a broader problem setting. Also, a scientific aim of the study could be formulated.

Clarified and better stated aims added.

2. Measurement site and general characteristics

RC1_9: P2, L29: I would avoid using the term "seasonal sea" which is not commonly used in the scientific literature.

Changed

RC1_10: P3, Fig. 1: It is quite empty. At least the sub-basins of the Baltic mentioned in the text should be shown (Archipelago Sea, Baltic Proper, Gulf of Finland, etc.). Also, consider presenting a local map where the oceanographic measurement site with the topography of the surrounding area could be seen.

Added to the map. We also added a local map around Utö in the appendix (Figure A2).

RC1_11: P4, L3-5: I agree that saline water inflows and freshwater input keep the stratification strong, but I do not agree that it causes the deep water to be anoxic. It could be opposite – saline water inflows could ventilate the deep layer. Moreover, the main reason for oxygen depletion is consumption of oxygen.

Removed

RC1_12: P4, L13-15: It is not obvious how the bottom topography and prevailing winds cause strong currents in the Utö Deep. Please, give a reference or explain it (also providing a map with topography if appropriate).

Finnish Geological Survey has mapped the areas close to Utö and found eroded areas at the bottom. We have access to this data, but are not able to publish it. The potential for strong flows is clearly visible in the canyon-like shape of the Utö deep (Figure A2).

In October 2017, we deployed an ADCP in the area, but unfortunately the acoustic releasers did not function, so we haven't been able to obtain the data yet (we try a ROV recovery later this year). We deployed a new ADCP in April 2018, so current data will become available in spring 2019.

3. Observations and methods
RC1_13: P4, L28-29: Please consider other options for sub-titles, e.g., "Observations and methods" instead of "Observations and methods used in this study"

Changed.

RC1_14: P6, L25-26: I do not understand the last sentence of this sub-chapter "In aim to keep the focus of this paper solid, we focus on a selection of the variables instead of all possible seasonal data." Please, rephrase it.

Removed. The aim is to simply state that with the available data set, we can calculate a lot of different things, but decided to focus on certain aspects only, to limit the length of this paper and keep it easier to read.

4. Results

RC1_15: P7, L9-13: Is the annual mean NAO index the best parameter to use here? For instance, Lehmann et al. (2011) did use NAO winter index (from December to March). Also, I do not see that the highest negative NAO values and low temperatures are connected (e.g., 2009 has far lowest NAO but not the lowest temperature).

We slightly modified the sentence as it was not accurate.

[Figure]

Figure R4: Annual average temperature as a function of annual average NAO. The average temperature for NAO < 0 is 5.7°C and for NAO>0 it is 6.3°C .

There is a lot of scattering in the data (Figure R4), but we see that the warmer years are typically with higher NAO index values and vice-versa. If we plot instead of annual average temperatures the winter time temperatures as function of winter time NAO, we see somewhat higher correlations.

RC1_16: P9, L9 and Fig. 6: Why median values are used here?

There was no particular reason to choose between mean and median. The resulting figure is not sensitive for this choice.

RC1_17: P9, L13-14: I do not agree with the suggestion that the relatively strong currents are the reason for the absence of the halocline. Could the bottom topography restrict saltier water transport into the Utö Deep?

We agree with the comment and have modified the article accordingly.

RC1_18: P11, L7-9: What do you mean by "we calculated the top and bottom of mixed layer depths"? I did not find a method in Lips et al. (2016) for that.

Reference left out and text modified.

What we actually did: We used a method inspired by the paper by Lips et al (2016) to determine the location of mixed layer. We basically described the top of the thermocline to be the location where the density had increased 0.25 kg/m3 from the density at 5 m and the bottom of thermocline the location where the density was 0.25 kg/m3 less than the density at 50 m. We also required the total density difference between 5 m and 50 m to be at least 0.5 kg/m3. Visually, this gave quite reasonable values. But this is now left out.

RC1_19: P11, L13-16: What could be these other phenomena responsible for deep water temperature increase in the 1980s and 1990s and recent decrease?

We do not know. In aim to see the potential effect of river runoff, we combined a data series for total Baltic Sea river runoff for the period 1900-2016. As we think these data will support the discussion in the paper, we included it as third panel in the Fig. 10.

We also added the following paragraph in the section 3.1 Observations and other data:

"River run-off data for the period 1900-2016 is a combination of observations for the period 1900-1995 (Hansson et al., 2011) and modeling for the period 1996-2016 (Johansson, 2018). The offset between the two data sets was corrected by calculating averages for both data sets for the overlapping period 1950-1995 and correcting the modeled data with the difference "

We also studied data by plotting e.g. monthly averages of sea water temperatures and salinities at different depths and tried to see if we can better figure out the differences. For example, we saw that during the period 1990-2000 the autumn time salinity difference between 90 and 50 meters was ~0.1-0.2 g/kg, while in 2013-16 it was almost 1 g/kg. As the the vertical mixing takes place during this period (Fig. 6), there is more resistant for the vertical mixing during the later period.

We will understand this phenomenon better in the future, when we have collected at least one year of high time-resolution data on sea water temperatures, salinities, underwater currents with the new instruments installed in April 2018.

RC1_20: P12, L3: Stratification should not be large and small, but rather strong and weak.

Corrected.

RC1_21: P12, L9-11: Do you explain the observed changes in deep water temperature by vertical mixing and stratification only, or is it possible that these changes are related to lateral

exchange? Please, explain it because it is not clear what are "these changes... responsible for the increased water temperatures at 50 m and 90 m depths".

There has not been significant changes in the wind pattern during the period 1960-2016, as seen from Fig. R5 below. Thus, while for individual periods the lateral exchange has definitely a strong impact, we do not assume it has influenced the trends.

RC1_22: P12, L14: What is meant by "We also see a rapid increase in 1940s"?

Added word "salinities"

RC1_23: P13, L7-8: Why these 30-year periods were selected for the comparison. It could be reasonable for the atmospheric data, but not for the oceanographic parameters which revealed a rapid change in the 1990s.

We decided to use the standard climate periods. However, we added a clarification in the text stating this.

RC1_24: P13, L14-15: Has this sentence ("However... ") the meaning of the previous comment that the chosen periods hide the rapid change after the 1980s?

Sentence improved

RC1_25: P14, Table 1: How these averages and standard deviations were estimated?

Simply calculated from the data using standard Matlab functions. We found a small inconsistency in our calculations and re-calculated the values and updated the table accordingly.

5. Conclusions

RC1_26: P15, L15: The saline water inflows increase salinity in the deep layer, but they could cause either an increase (mostly) or decrease in temperature. This change is not directly connected to the reduced vertical mixing. Do you have any pieces of evidence that the observed decrease in deep water temperature was due to the reduced vertical mixing?

If we look Figs 7 and 8, we see that there is larger salinity gradients in 1910-1920, 1960-1970 and again after the last increase (2013→) in salinity. In all cases, the bottom water temperatures have been lower as well. It is not of course possible to definitely say this is the case, but based on the data, we think is may be the case. Please see also reply to RC1_19.

**Referee 2, I. Vuorinen**

RC2_1: 1General comments:
The paper presents first time digitized and quality assured oceanographic data from the Northern Baltic proper in (semi) open sea conditions. Temporal range of the data is impressive, starting in 1881.  Spatially the data is from one spot, which lies somewhat mid-way between open sea and coastal conditions, also between the Bay of Bothnia and the Gulf of Finland. Approach and methods are basic, which is okay for this type of paper (presentation of a new, significant data set).

The authors thank for the support for publishing this paper.

RC2_2:The discussion could be somewhat more extensive (see below in specific comments). Presentation of figures and tables is appropriate (with some exceptions which are commented below), and English is mostly okay. I suggest below some places where wording should be reconsidered.

I suggest publication with minor corrections.

Specific comments
RC2_2: Title:  One hundred years of atmospheric and marine observations at Utö Island, the Baltic Sea.  -There are several islands with that name in the Baltic.  I know of one in Sweden and two in Finland.  Consider adding the coordinates, the country, or other information in the title in order to avoid mixing at least with the Utö in the Stockholm archipelago.

We agree with the comment especially with the possible confusion to Swedish island Utö and changed the title to "One hundred years of atmospheric and marine observations at Finnish Utö Island in the Baltic Sea"

RC2_3: Abstract:  I like the last sentence.  It points out a possible tipping into a new phase. This idea should be discussed more thoroughly, considering a possible breakpoint, its temporal  location  and  affecting  mechanisms.

Our aim in this paper was to show the long tradition of observations at Utö, a station that will be even more important multi-scientific observation site in the future. We wanted to give background for the future studies by showing some trends and by quantifying changes that can be seen from those observations. In addition to that, we pointed out some potentials of our data sets. We agree that the datasets raise interesting questions, that should be studied further. However, such analyzes would need e.g. support from climate and hydrodynamic models.

RC2_4: I agree,  that  would  be  obligatorily speculative, but as at present this idea seems to be the only one outcome suggested by authors,  it would be important to ponder it more deeply. I miss other conclusive sentences, such as what would be the best, or more appropriate way

(instead of just assuming a linear model) to analyze the evidently non-linear change over time which is seen in many parameters, such as in the salinity. I agree that the linear analysis should be the one to start with, but I also expect the authors to show capability to go further. Seeing abrupt changes like temperature since the 1980s and salinity at Utö makes me look for possible explanations and coincidences.  You could suggest a way forward, and the use of e.g. breakpoint equations in coming analyses, with other, non-linear, models.

We are using dynamical linear models (DLM) in the analysis, which does not correspond to traditional linear trend fitting. The trend estimated is not just a straight line but a function taking account the changes of the variables in time. See e.g. nice tutorial by Marko Laine in http://helios.fmi.fi/~lainema/dlm/dlmtut.html . DLM is actually a state-space model capable to model univariate or multivariate time series also in presence of non-stationarity, structural changes and irregular patterns.

RC2_5: Page 2 lines 10-12, you aim the paper to "analyse these time series in order to get information on typical atmospheric and marine conditions", but reading the paper makes me think that several less typical phenomena are shown, such as a rapid increase of salinity, or a decrease and disappearance of the ice cover and a subsequent suggestion of a shift of balance in the climate of Utö into a new phase. So I suggest rewording these lines, for the reader not expect too much of "just typical" happenings being observed.

We have now improved the introduction and better described the aims of the paper. Please see also replies to Referee 1

RC2_6: Line 31, you give the coordinates and write about the observation site and about the Island.  Are these coordinates for the midpoint of the Island or the lighthouse or coordinates for atmospheric observations?  Compare to page 4, line 33, where you give coordinates for the oceanographic sampling point.

Coordinates for all measurement site on Utö island and surrounding sea areas added to the caption of (new figure) A2 in the appendix.

RC2_7: Page 3, the map should have two panels, one showing the location in the Baltic sea (the present one) and another to show local bottom topography, depth etc.

A new figure A2 added to the paper.

RC2_8: Page 4, line 1, "with permanent pilots living on the island for generations" this is repetition of the information of the first part of the sentence, and, besides, "pilots living for generations" sounds improbable. Remove the sentence.

Corrected

RC2_9: Line 5, you write that: especially the deep samples may be considered to represent conditions of the Baltic Proper. On the other hand you write (page 9, line 1-2): we do not see any permanent halocline (and comment that possible cause to the halocline missing could be mixing due to currents.) These two statements are contradictory, first one is by Ahlnäs in 1961. Have the circumstances changed between then and now? Lack of the deep halocline also puts the sampling station oceanographically more to the Bothnian Bay side than on the Northern Baltic Sea. Could you comment on that?

This is a good point, please see the detailed reply to comments by Referee 1. This discussion is also now reflected in the paper.

RC2_10: Line 7. I do not accept the phrase about biological characteristics. First: there is no information included in writing that "biological characteristics are typical for the outer archipelago" as this is anyway the basic assumption. Secondly, this kind of basic assumption is not valid for this location as biological characteristics point out to a eutrophic environment. Since the 1980′ s the cyanobacterial blooms have been observed in this area, but before that the area, as the whole Northern Baltic Proper was considered to be an oligotrophic environment. Same rapid change from oligotrophy to eutrophy is seen in, for example, in Secchi disc readings in the Gulf of Finland during the 1900. Please give appropriate information on biological change over at least the last decades, as you do for the sea ice in the next sentence.

We decided to remove the comment on typical biological characteristics None of the authors is a marine biologist / limnologist and we do not feel competent to discuss the biological characteristics on scientific level. We have a publication focusing on biological aspects in preparations and we will include better description of local ecosystem in that paper,

RC2_11: Page 6, lines 32-33, you write that you investigated the annual average temperatures against the NAO indices in Fig 3., but that figure only shows the NAO history, while temperatures are given in the Fig. 2. You also claim finding, for individual years, lower temperatures connected to highest negative NAO values, but in Fig. 2 there are no temperature values given for individual years at all. You refer also to Fig. 2 having lowest temperature values (5 year periods) in around 1980, while this period (1980) in the NAO figures just show mid values of the index. What are "highest negative NAO values" anyway? Are they just lowest values of the index, or something else? Rewrite this part.

Please see the comments to the referee 1 above. We improved the text for this part.

RC2_12: Page 7, line 9 and 12, you write about manual observations, you should write about visual observations.

We added word visual and put word "manual" in parenthesis. This is terminological issue, we normally call man-made observations "manual" and machine-made "automatic".

RC2_13: Page 8, line 3, you mention not to have found significant changes in wind speed. Okay, but my personal observation from Utö station when comparing wind observations before and after the 1970s was that there was a substantial reduction of completely calm days (see attached figure which is based in Finnish Meteorological Institute observations at the Island of Utö)). So the overall windiness has increased anyway. As you suggest, more analyses are needed. You could try and include also the data on calm days.

We calculated histogram of wind speed and directions (Fig. R5) for different periods and found no significant changes in wind directions or wind speeds. We do not have data prior to 1960 properly quality assured yet, so unfortunately it is not possible to see any longer term changes.

[Figure]

Figure R5:  Histogram of wind directions and speeds for the period 1960-2016 and three shorter periods, 1960-1979, 1980-1999 and 2000-2016.

RC2_13: Page 9, line 4-5, remove the sentence: As our focus is..”  and start directly from: We decided to... Lines 14-15 you write that:  “the surface temperature follows the ... .atmospheric temperatures Fig. 2) ... with a rapid increase in 1980s,  which is okay and correct, but then you write that: “and a warmer period from 1930 until 1960s” which, however is not seen at all in the Fig 2 which you refer to. Rewrite that part.

Beginning of the sentence removed. We do feel that there is a period with higher temperatures during this period, ~1930-1960 and it is visible in the top panel of Fig. 2

RC2_14:  Page 13, line 10, "reduces the lowest temperatures" sounds strange. Consider rewording.

.. since open sea is a large source of latent heat, which leads to higher atmospheric temperatures than when the sea is ice covered

RC2_15:  Typos: page 9, line 4 reads: one hundred year, should read one hundred years. Please also note the supplement to this comment:

Corrected

[revised manuscript text omitted]